# Bone Healing Gone Wrong: Pathological Fracture Healing and Non-Unions—Overview of Basic and Clinical Aspects and Systematic Review of Risk Factors

**DOI:** 10.3390/bioengineering10010085

**Published:** 2023-01-09

**Authors:** Dominik Saul, Maximilian M. Menger, Sabrina Ehnert, Andreas K. Nüssler, Tina Histing, Matthias W. Laschke

**Affiliations:** 1Department of Trauma and Reconstructive Surgery, Eberhard Karls University Tübingen, BG Trauma Center Tübingen, 72076 Tübingen, Germany; 2Kogod Center on Aging and Division of Endocrinology, Mayo Clinic, Rochester, MN 55905, USA; 3Institute for Clinical and Experimental Surgery, Saarland University, 66421 Homburg, Germany

**Keywords:** bone, bone healing, fracture, non-union, pseudarthrosis

## Abstract

Bone healing is a multifarious process involving mesenchymal stem cells, osteoprogenitor cells, macrophages, osteoblasts and -clasts, and chondrocytes to restore the osseous tissue. Particularly in long bones including the tibia, clavicle, humerus and femur, this process fails in 2–10% of all fractures, with devastating effects for the patient and the healthcare system. Underlying reasons for this failure are manifold, from lack of biomechanical stability to impaired biological host conditions and wound-immanent intricacies. In this review, we describe the cellular components involved in impaired bone healing and how they interfere with the delicately orchestrated processes of bone repair and formation. We subsequently outline and weigh the risk factors for the development of non-unions that have been established in the literature. Therapeutic prospects are illustrated and put into clinical perspective, before the applicability of biomarkers is finally discussed.

## 1. Introduction

A broken bone starts to heal at the moment of fracture. The bone tends to re-unite in either a direct or indirect manner through callus formation, and with or without surgical help [1]. Since Paleolithic times, from which conservative treatment of Lisfranc’s fracture has been reported [2], to Neolithic cranial surgery [3] and evidence of bone remodeling after surgical intervention in Medieval times [4], to modern minimally invasive spine [5] and robotic trauma surgery [6], the aim of successful bone healing has been perpetually pursued. However, it is not always achieved. In fact, based on the definition and localization, non-unions occur in every 10th to 20th fracture [7].

According to the AO principles, a non-union is a fracture that has not healed after 6 months. This definition is used by most surgeons in daily practice [8,9]. As defined by the U.S. Food and Drug Administration (FDA) in 1986, non-unions of diaphyseal fractures are indicated by a “failure to heal at 9 months with no progress during the previous 3 months” [8]. Bony non-union must be distinguished from pseudarthrosis, where the non-union is present for a long time, and the bone ends are sclerotic, forming a synovial articulation [8]. Among non-unions, hypertrophic (usually hypervascular and vital) and atrophic (avascular and non-vital) non-unions represent two biologically distinct phenotypes [10].

It should be noted that primary bone healing and secondary bone healing occur in different fracture types. Primary bone healing is achieved by rigid stabilization leading to interfragmentary compression and minimal motion of the fracture (<0.15 mm contact zones) [11]. In consequence, no callus is formed. In secondary bone healing, achieved by a nailing approach or cast/brace therapy, callus is formed due to larger interfragmentary movements (0.2–1 mm contact zones) [12]. 

This review summarizes the epidemiological aspects of fractured bones and the socioeconomic consequences of non-union occurrence. Subsequently, the key cellular players in the (non-)physiological healing process are described and their disturbance briefly discussed. These cellular insights are followed by a systematic review of the risk factors for non-union. Finally, therapeutic options are outlined and the application of potential biomarkers discussed.

## 2. Materials and Methods

### 2.1. Search Strategy for Systematic Review

For the systematic review of the risk factors for non-union, the Medline, Cochrane, and Google Scholar databases were consulted. Original articles, meta-analyses, and reviews were selected and considered. Following keywords were included: “pseudarthrosis”, “non-union”, “delayed healing”, and the Boolean operator “AND” combined with “bone healing” or “fracture” or “fracture healing”. Abstracts from the earliest available records until November 2022 were considered, and reference lists of original articles and reviews were manually evaluated to identify additional relevant studies. The respective algorithm is demonstrated in Figure 1. The Preferred Reporting Items for Systematic Reviews and Meta-Analyses (PRISMA) statement and checklist were used throughout this review [13].

### 2.2. Eligibility

This study followed the participants, interventions, comparisons, outcomes, and study design (PICOS) framework [14].

### 2.3. Population

The population of interest were humans who had experienced a fracture.

### 2.4. Intervention (Exposure)

The exposure was fracture treatment, either conservatively or operatively, in both the study group and control group.

### 2.5. Comparison

Comparators were considered throughout the review, i.e., smoking status or fracture characteristics.

### 2.6. Outcome

The outcome parameters were the time until complete healing or time until non-union was declared.

### 2.7. Study Design

All studies were eligible, with the exception of narrative syntheses, case studies or studies not available in the English language or as a full text.

### 2.8. Inclusion Criteria

From 3017 results and after screening all headings, 626 abstracts remained, out of which 23 original articles and 5 reviews were included.

### 2.9. Exclusion Criteria

Studies were excluded if their manuscripts were not available as full text, as were articles exclusively containing animal studies. Biomechanical studies were also excluded.

### 2.10. Search Strategy for Narrative Review

For the narrative review particularly focusing on the cellular elements contributing to (impaired) fracture healing, additional animal studies were included due to a lack of human data.

## 3. Epidemiology

Non-unions appear rarely, but represent one of the most severe complications in trauma and orthopedic surgery. Their frequency depends on the applied definition and varies between 2–10% of all fractures, but may reach 50% in open tibial fractures [7,15,16]. Delayed healing (no bony healing after four months) [17] affects an estimated 600,000 fractures per year, and around 100,000 fractures in the US display non-union [12,18]. International data are limited, however, the global incidence of non-union is estimated at around 5–10%, depending on the fracture location and country, with China reporting a non-union rate of 4.7% in tibial fractures, and Singapore reporting a 42.7% non-union rate (only open Gustilo–Anderson IIIB tibial fractures were included). Smaller unrepresentative studies in Turkey and Egypt reported lower rates of tibial non-unions at 1.4 and 3.3%, respectively [19].

The highest rates of non-unions were observed in the tibia, clavicle, humerus and femur, and interestingly were not more frequent in older populations [7,20].

Patients with non-union suffer from tremendous secondary functional deficit and are frequently unable to participate in daily life, and substantially benefit from a structured therapeutic approach taking into account the occurrence of infection, impaired biology, and concurrent metabolic disorders [21]. An effective therapeutic approach is based on the application of growth factors, scaffolds, and mesenchymal stem cells (MSCs) into the fracture gap, while the mechanical environment is stabilized, designated as the “diamond concept”. A more detailed characterization of this therapeutic option is described below [22,23].

The approximate costs for treatment of long-bone non-unions are estimated to be at least USD 11,333 in the US [12], USD 17,000–18,000 in the UK [24], USD 9641 in Australia [25] and between USD 2900 and USD 6300 in Germany [16]. In the USA, a tibial shaft non-union was USD 13,870 more expensive than the regular healing process [26], while overall costs of delayed fracture healing in the USA are estimated at around USD 14.6 million annually [15]. Indirect costs of productivity loss are difficult to calculate, but even a delay of surgery from <12 to >12 h after injury may lead to additional costs of USD 5520 per case [12,27].

## 4. Cellular Components Contributing to (Impaired) Bone Healing

The fracture process can be subdivided into four intergradient stages. The initial stage of fracture hematoma and inflammation is followed by the second stage of cartilage formation with angiogenesis (soft callus). The third stage is characterized by cartilage removal and calcification (hard callus), before the fourth stage of chronic bone remodeling concludes the healing [28]. The complex interplay of MSCs with local macrophages, osteoprogenitor cells, osteoblasts, osteoclasts, and chondrocytes is crucial to ensure a regular healing process (Figure 2). Numerous conditions including smoking and disease-specific peculiarities have been found to impair this well-balanced process [28,29,30,31]. In order to reveal the cellular basis of impairment in non-unions, individual cell types are here illustrated separately before focusing on the microenvironment.

### 4.1. MSCs

Undifferentiated MSCs display a “diamond” component, representing the osteogenic origin of local proliferation in the fracture callus [11]. Intriguingly, while MSCs in non-unions were found comparable to MSCs in unions in terms of proliferative capacity and cellular viability [36,37], the serum of donors with non-unions within one week after fracture negatively affected MSC proliferation [38].

As expected, aging appears to impact the total number of MSCs available at the fracture site and their differentiation potential. While the total number of MSCs harvested was significantly higher in aged mice, the number of colonies formed was substantially lower compared with young animals [39]. In human iliac crests, however, a decline in the number of precursor cells has been shown to start as early as the second decade [40]. In human non-union tissues, increased levels of senescence-associated beta galactosidase (SA-β-Gal) activity have been found in MSCs, potentially indicating a senescent cell state in non-union bones [41].

The efficiency of MSC transplantation in the treatment of bone non-union has not been clinically validated. Although four clinical trials have been completed, none have yet published their results (NCT02177565, completed in October 2011, NCT01206179, completed in May 2011, NCT02230514, completed in December 2019, NCT01788059, completed in 11/2013). Nonetheless, bone grafting with either autograft or allograft containing MSCs (of undefined proportions and quality) remains the current gold standard in patients with incomplete bone healing [42].

A cell-free approach to treatment of bone defects involves MSC-derived extracellular vesicles (MSC-EVs) [43]. These may contain monocyte chemotactic protein 1 (MCP-1) and several angiogenic factors that accelerate fracture healing in rodent models [44,45,46].

Despite several experimental approaches, the exact mechanisms by which MSCs interfere with the fracture-healing process are poorly understood. In addition, cellular processes and components involved in fracture healing are yet to be elucidated in detail [28]. Inconsistent terminology regarding the use of stem cells as therapeutic tools led to development of the DOSES (donor, origin tissue, separation method, exhibited cell characteristics associated with behavior, and site of delivery) concept to ensure that future research is reproducible [47].

### 4.2. Macrophages

Macrophages are myeloid lineage cells, differentiated from hematopoietic stem cells, that reside in periosteal and endosteal tissues and are frequently referred to as osteomacs [48]. M1 macrophages participate in the inflammatory phase of fracture healing. Although their amplified activation resulted in compromised healing in rats [49], low-dose TNF-α administration accelerated healing in a murine model [50]. Macrophages are key osteoinductive mediators through their secretion of cytokines including interleukin 1 (IL1), IL6, IL12, and TNF-α [11]. Moreover, reduced numbers and impaired function of M2 macrophages (alternatively activated via cytokine exposure, as opposed to M1 macrophages which are classically activated via interferon gamma (INF-γ) or lipopolysaccharides) have been linked to reduced vascularization in older rats [51]. Macrophages produce vascular endothelial growth factor (VEGF) [52,53] and subsequently promote vascular sprouting at the fracture side [48], an effect that may decline with age [54].

### 4.3. Osteoprogenitor Cells

Osteogenic cells arise from the local periosteum and potentially to a small proportion from C-X-C chemokine receptor type 4 (CXCR4)-positive circulating osteoprogenitors [32]. In a small case study of humans, large bone defects (≥4 cm) were filled with osteoprogenitor cells grown on scaffolds in order to achieve bony union [55]. Interestingly, the differentiation capacity of osteoprogenitor cells was affected by trauma hemorrhage, corresponding with the clinical observation of non-union frequently occurring in polytraumatized patients [56,57,58]. It has been shown that osteogenic extracellular vesicles (EVs) originating from injured brain tissue target osteoprogenitors and accelerate bone healing [59]. 

A decline of progenitor cells in bone marrow aspirates with age has been demonstrated in women, but not in men [60]. Interestingly, a lower level of progenitor cells was also detected in the iliac crests of patients with non-union compared with control patients [61].

While NOTCH signaling is dispensable in both osteoblasts and chondrocytes, it is imperative in osteoprogenitors to enable normal bone healing [33].

### 4.4. Osteoblasts 

Osteoblasts are bone-synthesizing cells originating from MSCs. They appear to be particularly important in the middle stages of bone healing. Osteoblasts have been found to be unaltered in the early stages of non-union development, while in observations of the osteoblastic maturation process after 5 and 10 weeks, runt-related transcription factor 2 (RUNX2)- and osteocalcin (OCN)-positive cells were significantly downregulated in a murine atrophic non-union model [62]. In addition to the total number of cells, their viability may be a reason for the occurrence of non-unions; osteoblast cell viability and the signaling of Wnt, insulin-like growth factor (IGF), transforming growth factor (TGF)-β, and fibroblast growth factor (FGF) were significantly downregulated in fracture non-unions compared to healthy endosteal sites [34]. In line with these findings, IGF1 deficiency has been demonstrated to impair endochondral bone formation in fracture healing by impacting osteoblast differentiation and coordination of chondrocyte, osteoclast, and endothelial cells [63].

### 4.5. Osteocytes

Osteocytes are mechanosensitive cells that respond to the deformation of surrounding tissue via flow-induced shear stress on their surfaces. Their exact function and the role of the lacuno-canalicular network has been outlined in detail elsewhere [64]. These mesenchymal-derived cells are most abundant in adult bone, where they coordinate osteoblast and osteoclast activity and arrange adaption to environmental changes via release of various molecules [65,66]. While their exact role in fracture healing is largely unknown, their involvement in aging and senescent phenotype has been linked to age-dependent skeletal decline [67,68].

### 4.6. Osteoclasts

Osteoclast activity was markedly increased in human scaphoid non-unions, and expanded osteoclastogenesis may be one factor leading to non-union [69]. While osteoclast–MSC crosstalk stimulates both the recruitment and differentiation of osteoblasts, pre-osteoclastic cells were predicted to accelerate fracture healing [70]. In a murine atrophic non-union model, enhanced osteoclast activity was observed, together with increased TGF-β, TNF-α, and receptor activator of nuclear factor kappa-Β ligand (RANKL) levels, particularly in the early stages of non-union [62].

### 4.7. Chondrocytes

Chondrocytes are hypothesized to transform into osteoblasts, as demonstrated in Agc1CreERT::Ai9 mice, in which SRY (sex determining region Y)-box 2 (SOX2), octamer-binding transcription factor 4 (OCT4), and NANOG were co-stained in chondrocyte-stem-cell-like cells. This mechanism may be perturbed in non-unions [71,72]. In fact, chondrocyte trans-differentiation may be the primary path of endochondral bone formation in bone repair [72]. Delayed chondrocyte hypertrophy was shown to impair bone formation, potentially through over-expression of SMAD family member 6 (SMAD6) [73]. Interestingly, age-related decline in chondrocyte activity along with increased cellular senescence and reduced responsiveness has been detected in human articular cartilage chondrocytes. However, it remains unknown whether this is applicable to the fracture callus [74].

### 4.8. Vasculature

The importance of vascularization for the healing fracture has been highlighted in several studies [35,75,76]. Indeed, microparticle-delivered VEGF and bone morphogenetic proteins (BMP)-2 improved bone healing in murine atrophic non-unions, evidenced by enhanced stiffness and bone volume within the callus [35].

In hypertrophic non-unions, the density of blood vessels was comparable to that in regular healing bone in a human study of non-union vs. healed fractures, in scaphoid non-unions, and in animal models [69,77,78]. While MSCs isolated from non-union sites showed similar differentiation capacity, they did not inhibit early-stage in vitro angiogenesis to the same extent as their periosteal counterparts, potentially participating in an immature vascular network [36]. The definite function of vasculature and its role in non-unions subsequently warrants further elucidation.

Overall, the roles of several key cellular players within the healing bone have been well studied. However, the mechanistic details of non-union, its cellular origins, and related disturbances remain largely unknown to date. With novel techniques such as single-cell RNA sequencing (scRNA-Seq) and the broad applicability of spatial transcriptomics, major progress is to be expected in the field of bone (non-)healing in the coming decades [79,80,81].

## 5. Risk Factors for Non-Union

Numerous studies have analyzed risk factors for bone non-union. The following systematic review (see also Figure 1) evaluates the available literature relating to human bone non-union, and distinguishes retrospective from prospective studies and meta-analyses (Figure 3, Table 1). The established risk factors for non-union formation can be subdivided into influenceable and irreversible hazards, patient-dependent or patient-independent jeopardies [18]. Additional complexity is added by considering biological and surgical risk factors. Controlling for all these factors would make a prospective randomized controlled trial (RCT) unfeasible, requiring an unrealistic number of patients [82]. However, certain factors have been examined with rigor, while others remain suggestions without scientific evidence (Figure 3) [7]. The studies included herein are summarized in Table 1.

Age itself may present a risk factor for non-unions in the clavicle [83,84,86], however the literature for the humerus is indecisive [85,87]. A recent meta-analysis was unable to include age, due to the heterogeneity of data presentation [88].

A larger systematic meta-analysis concluded sex not to be a risk factor for non-union (odds ratio [OR]: 1.0) [88].

Smoking has been validated as a risk factor for non-union in the clavicle [92], diaphyseal fractures in general [93], and tibial fractures in particular [89,90].A systematic review and meta-analysis calculated the OR of non-union in smokers to be 2.32 in general and 2.16 in tibial fractures [97], while another systematic review and meta-analysis calculated a similar OR of 2.5 for non-union [98]. Importantly, a control for smoking-related risk factors (i.e., further diseases) might reduce the calculated OR for smoking as an independent risk factor [82]. Another systematic review and meta-analysis pooled 7516 procedures to reveal a 2.2 times higher risk of delayed healing or non-union in smokers [99]. A meta-analysis pooled 38,465 patients, and found that smoking status led to an OR of 1.7 for non-union [88]. A prospective trial including 647 patients calculated an OR of 2.4 for the development of tibial non-union in smokers [91]. In a large retrospective study of proximal humeral fractures (2230 patients, age > 18 yrs), smoking was independently predictive of non-union according to multivariate analysis [94].Therefore, cessation of smoking and limiting the use of other drug treatments such as NSAIDS is recommended to improve chances of fracture healing [110].

Open fractures have been associated with higher rates of non-union [90]. In a systematic review and meta-analysis, the OR of non-union in open fractures was calculated to be 1.95 [97]. In another retrospective case-control study, an OR of 2.71 was measured for open fractures leading to non-union [100]. In a retrospective study (486 patients, 56 non-unions), open fractures had significantly higher risk of leading to non-union in fractures of Gustilo type IIIb or higher (OR 4.91 [101]).

Displacement has been analyzed in fractures of the clavicle. Here, displaced fractures are a predictive factor for non-union [84]. In a prospective study of 222 conservatively treated patients, non-union of clavicle fractures occurred in 15 individuals, and displacement (> one bone width) predicted complications but not necessarily non-unions [102]. In another prospective trial with 941 patients receiving non-operative treatment for clavicle fracture, fracture displacement was mildly predictive for non-union formation [92].

Diabetes appears to be a risk factor for non-union, as shown in a meta-analysis [88]. However, larger randomized controlled trials have been surprisingly scarce.

Obesity has been identified as a risk factor for non-union even in pediatric patients [103,104], and was assessed with an OR of 1.5 for non-union according to a meta-analysis [88].

When osteoporosis is included in the list of risk factors for non-union, evidence is conflicting. Despite some evidence of osteoporosis (albeit not independently) negatively affecting bone union in according to univariate analysis [95], a prospective trial with 1498 patients after 1:2 matching indicated that bone mineral density (BMD) did not predict the rate of non-union [105]. However, a retrospective case-control study revealed an OR of 3.16 for osteoporosis as a contributory factor to non-union [100].

Literature on the consumption of alcohol and its effect on fracture healing is inconsistent. In a prospective trial involving 122 patients with femoral neck fractures, alcohol was a risk factor according to a multivariate analysis [106], although a systematic review and meta-analysis detected no differences in alcohol drinkers vs. non-drinkers in terms of bone non-union [98].

Drug treatment may also intervene with physiological bone healing. The most prominent drugs in this respect are NSAIDs and steroids. NSAIDs have been frequently discussed as a risk factor for non-union. In a large matched-control study, administration of NSAIDs 12 months before fracture led to an OR of 2.6 for fracture non-union [107], while a smaller retrospective case-control study found a strong association between delayed healing and the use of NSAIDs before injury [96]. In a retrospective case–control study, NSAIDs were associated with an OR of 2.04 for non-union [100]. Likewise, steroids are frequently listed as risk factors for non-union, although data supporting this suggestion are lacking. In a prospective trial, steroid use did not yield a higher number of non-unions compared to the control group [91]. However, the use of reproductive steroids in pediatric patients may increase the risk of non-union [108]. Interestingly, data from animal experiment are inconclusive on whether corticosteroids negatively impact bone healing, and dose dependency is discussed [111]. Meanwhile, proton pump inhibitors (PPIs) have recently been suspected to increase the risk of non-union in femoral and tibial shaft fractures. A retrospective case–control study revealed any use of PPIs was associated with an OR of 4.5 for further risk of surgery due to non-union. However, the power of that study was reduced by its inclusion of a low number of patients with a non-union, their aggregation with delayed unions, and the retrospective nature of the study itself [109].

Immune dysregulation has been hypothesized to impair fracture healing. When the initial inflammatory phase and local immunosuppression are efficiently resolved, later stages require immune cells to guide MSC differentiation and activity [112]. In all healing stages, immune homeostasis is controlled by CD4^+^, CD8^+^, and regulatory T cells [113]. Accordingly, a T-cell defect or compromised immune system, as seen in HIV patients with disturbed levels of TNF-α, is associated with impaired fracture healing [114]. The complex interplay of inflammation and the immune system can be further affected by diseases such as diabetes, habits such as smoking, or even aging itself, and larger prospective trials on the effects of immune dysregulation on fracture healing are lacking [115].

## 6. Therapeutic Options

To treat bone non-unions, classical principles (biomedical approaches, i.e., via reamed stem cell transplantation, ultrasound, etc.) and novel approaches (biomaterials and nanotechnologies, genetic modification, etc.) can be applied.

The administration of BMP-2, BMP-7, platelet-rich plasma (PRP), and MSCs has been demonstrated in a recent meta-analysis to reduce time taken for bone repair [116]. As an elegant approach to accelerate fracture healing, stem cell therapy has been suggested, with the iliac crest being the most common and safest source of cells [117]. More locally, intramedullary reamer–irrigator aspirations (RIA), especially in the long tibia and femur bones, represent easy and reliable sources of stem cells [25,118,119]. Interestingly, without attempts to concentrate the number of colony-forming units, the number of real MSCs among hematopoietic progenitors and stromal cells may be low. The quantity was found to correlate positively with the volume of mineralized callus, according to a study including 60 patients with tibial shaft non-union [120]. Subsequently, a concentration process is needed [121,122].

Another method for the treatment of delayed healing or non-unions is low-intensity pulsed ultrasound (LIPUS), which in 1994 was approved by the FDA for acceleration of bone healing [123]. Despite initially positive smaller studies indicating a beneficial effect on fracture healing, particularly in tibial shaft and distal radius fractures [124,125], the TRUST study—a randomized, blinded, sham-controlled clinical trial—did not reveal any differences between LIPUS or a sham device in terms of outcomes or time to radiographic healing of tibial shaft fractures, which led to recommendations against the use of LIPUS in general [126,127]. Today, although several animal studies have provided favorable data for specific indications such as the healing of the bone-tendon transition [128], LIPUS is generally not used clinically except for rare individual treatments if surgery is not feasible [129].

Pulsed electromagnetic field stimulation (PEMFs) represents another approach for the treatment of bone non-unions, and is FDA-approved [130,131]. Preliminary studies are promising [132], with in vitro and in vivo trials supporting the use of PEMF in cases of osteoporosis and delayed union [133,134,135,136]. A clinical trial involving distal radius fractures is currently ongoing (“not yet recruiting” in 02/2020).

Biomaterial-supported therapies for bone regeneration include various different approaches. Among these, bone-graft substitutes including a combination of bio-active molecules and cell-based supplements with osteoprogenitors have been applied in vivo [42]. Bone-graft substitutes include autologous and allogenous bone substitutes. The former are limited in their availability from the iliac crest, and the latter are generally preserved in a freezing medium and consequentially lack some of the osteogenic properties and mechanical stability of autografts. However, the cryopreservation process avoids a donor-site immune response [137]. Despite these limitations, the implantation of autograft and allograft substitutes remains the current gold standard for treatment of larger bone defects [42]. The addition of bioactive molecules, such as growth factors BMP-2 or -7, PDGF or the peptide P-15, has been utilized in recent decades, supported by FDA approval for these [138]. Dose-dependent inflammatory side effects have limited the use of BMP-2 therapy [139], and results from new clinical trials on BMP-2 are expected soon (NCT02924571, NCT05065684, NCT05238740), while clinical studies on P-15 in bone grafting for spinal deformity (NCT05038527, recruiting in 02/2022) and lumbar fusion (NCT03438747, active, not recruiting in 02/2022) are ongoing. Cell-based supplements usually contain bone marrow stromal cells (BMSCs), adipose-derived mesenchymal stem cells (AMSCs), or periosteum-derived stem cells [42]. The adipose tissue is an additional and easily applicable source for MSCs, particularly for larger defects is, and can be efficiently harvested for the isolation of AMSCs [140]. BMSCs have been widely used in clinical trials, either with or without the addition of BMP-2, injected into a medium or seeded into a matrix before application in a bone defect. A clinical trial on intervertebral disc-defect repair with BMSCs on a gelatin sponge is currently recruiting (NCT03002207, 10/2021). A recent meta-analysis concluded that use of a scaffold positively impacts the healing rate of a cell-based treatment [141]. However, there remains a lack of prospective randomized controlled trials with long-term follow-up assessing scaffold-based BMSC treatment of non-unions. In addition to embedding freshly isolated MSCs into a scaffold, these can be further genetically modified in order to improve their osteogenic properties or proliferative potential. This can be achieved by viral-based gene-delivery systems using a gene-activated matrix, genome editing via the CRISPR/Cas9 technique, or engineered extracellular vesicles (EVs) [142].

The latter approach also represents a cell-free alternative. MSC-derived EVs alone have been shown to improve fracture healing in rodent models [44,46].

The convergence of mechanical and biological considerations, in addition to adequate vascular supply at the healing site and the physiological host state in the “diamond concept”, conceptualizes a framework of fracture repair comparable to a polychemotherapy attack on non-unions. Within this context, the healing process is facilitated by osteoinductive mediators (cytokines released by macrophages), an underlying matrix (extracellular matrix promoting “homing” and migration of osteogenic cells), osteogenic cells (i.e., multipotent MSCs and committed osteoprogenitors from the adjacent periosteum), and mechanical stability (sensed by osteocytes) [143,144]. In addition, adequate blood supply is necessary (periosteal vascularization is essential for the influx of osteoprogenitors and association of invading blood vessels with osteoblast precursors) [145,146] and beneficial host factors (no smoking, cessation of harmful drugs) may be involved [11,22,23].

## 7. Biomarkers for Non-Union

TGF-β1 has been described as a potential biomarker for non-union since a decline was detected in patients with diaphyseal long bone fractures [147], which was tested in two studies including 103 and 30 patients, respectively. In these studies, a decline of TGF-β1 occurred earlier in delayed union patients [148,149]. Newer data suggest that TGF-β1 impairs MSC-mediated bone regeneration mechanistically through BMP2 inhibition, but its role as biomarker has not recently been explored and no clinical trials are ongoing [150].

Although serum of non-union patients has been shown to impair bone marrow (BM)-MSC proliferation in vitro, the underlying factors were not identified, and cytokine levels were found to be similar in non-union and union serum samples [38].

However, pre-clinical studies have been promising. In a rat critical size defect model, cytokines including IL-10 and myeloid-derived suppressor cells (MDSCs) were significantly elevated in blood one week after treatment, prior to poor bone healing outcomes. Interestingly, blood B-cell levels were positively correlated to bone volume as early as one week after treatment, indicating an early predictor for successful healing. Given the importance of B-cell maturation for bone healing, these cells might be promising markers for bone repair, although clinical trials remain lacking [151]. Moreover, local bone marrow showed MDSC levels consistent with the blood immune-response profile 12 weeks after treatment. This promising cell population secretes various anti-inflammatory factors such as IL-10 and TGF-β, inhibits the immune response, and expands during aging [152,153]. Nonetheless, serum parameters to reliably predict non-union at the time of fracture have not yet been clinically confirmed [154].

In a small study (six unions and six non-unions, with half of the patients in each group having type 2 diabetes mellitus), Annexin A3 (ANXA3) was found to be significantly upregulated in blood samples of patients with non-unions compared with fracture unions [155]. As indicated in a study from 2020, matrix metalloproteinases such as matrix metalloproteinase 9 (MMP9) might be detectable in higher levels in the mononuclear blood cells (PBMCs) of patients with unions compared with non-unions (39 union vs. 16 non-union patients) [156]. Additionally, in another study from 2020, levels of placenta growth factor (PlGF) in the serum of patients with femur or tibia non-unions were significantly higher than those of union patients on days 1 and 3 (10 unions vs. 5 non-unions) [157].

These findings indicate that no reliable biomarker for non-union has been identified that is sufficiently understood and clinically confirmed.

## 8. Conclusions

Bone non-union occurs in 2–10% of all fractures. Although the cellular components of bone healing are partially understood, the mechanisms by which bone fails to heal are manifold and remain largely unexplored. Among the main risk factors for non-union are smoking, open fractures, and certain medications including NSAIDs. However, the literature lacks prospective randomized controlled trials to verify the results of an abundance of retrospective studies. The diamond concept has proven itself over the years, providing a six-sided convergence of considerations leading to non-union, and offering a guide to its treatment. Unfortunately, reliable biomarkers for non-union are scarce. Non-union remains a pivotal challenge for trauma surgery and orthopedics, and uncovering the underlying pathological mechanisms of non-union in order to develop new therapeutic options represents a major challenge for researchers and surgeons in the 21st century.

## Figures and Tables

**Figure 1 bioengineering-10-00085-f001:**
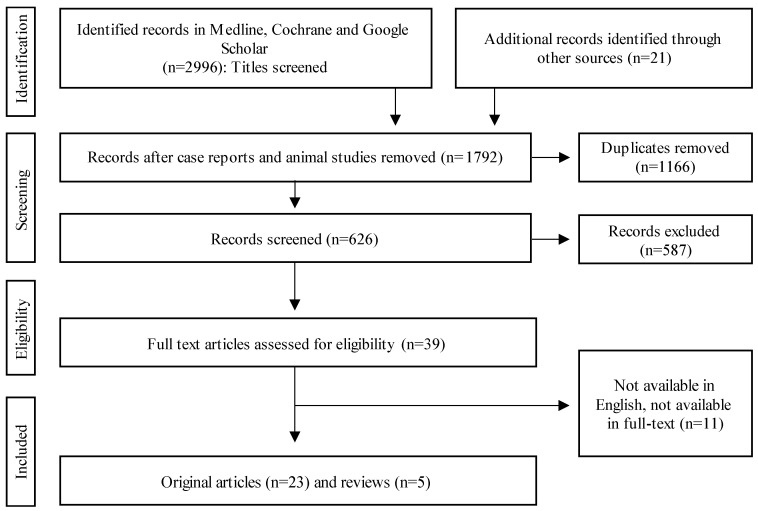
PRISMA flowchart of data extraction.

**Figure 2 bioengineering-10-00085-f002:**
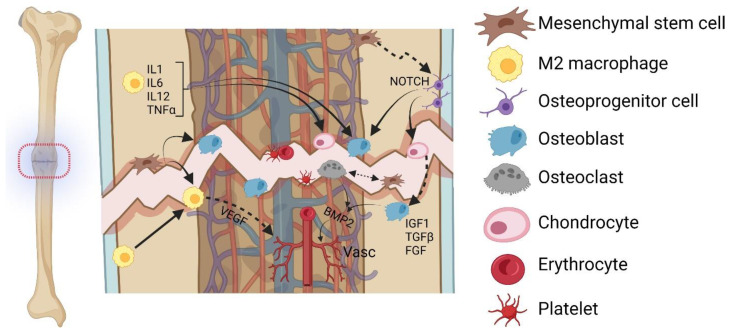
**Cellular components contributing to (impaired) bone healing**. The initial fracture gap is filled quickly by a hematoma, followed by the immigration of mesenchymal stem cells originating from the underlying periosteal layer [32]. These differentiate into osteoprogenitor cells (crucial neurogenic locus notch homolog protein 1 [NOTCH] signaling) [33], further giving rise to cells that clear the fracture site (osteoclasts) and form a fibrocartilaginous callus (chondrocytes, osteoblasts) before the inferior cartilage is transformed into ossified bone. Osteoblastic insulin-like growth factor 1 (IGF1), transforming growth factor β (TGF-β), and fibroblast growth factor (FGF) are decisive for bone union [34]. M2 macrophages orchestrate the healing process via interleukin-1 (IL1), IL6, IL12, and tumor necrosis factor α (TNF-α) [11]. The newly developing vasculature is stimulated by vascular endothelial growth factor (VEGF) and bone morphogenetic protein 2 (BMP2) [35], and erythrocytes as well as platelets form the initial clot in which the callus is formed. Created with BioRender.com.

**Figure 3 bioengineering-10-00085-f003:**
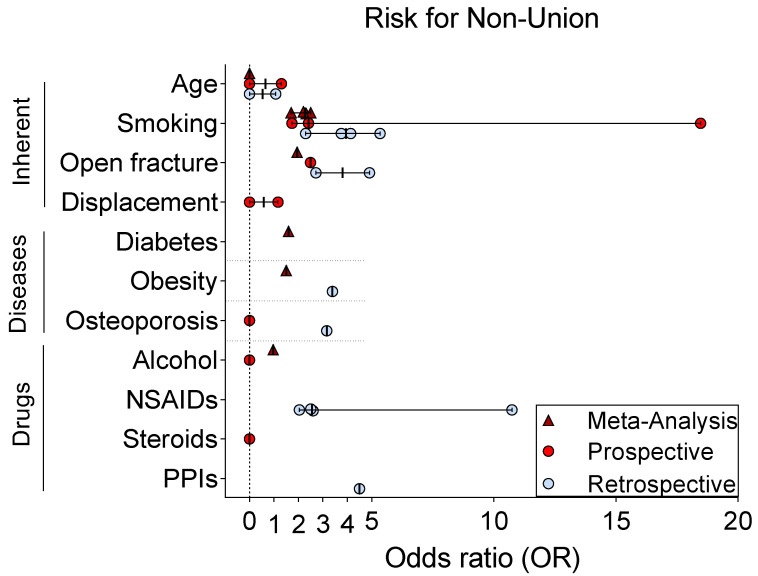
**Risk factors for bone non-union formation.** The plethora of studies on risk factors eliciting bone non-union makes comparison difficult. Many clinical studies focusing on factors associated with bone healing and related prohibitive diseases have reported that smoking had a harmful effect on bone healing. All study types indicate increased risk of non-union in open fractures. However, for non-steroidal anti-inflammatory drugs (NSAIDs) and proton pump inhibitors (PPI) only a few retrospective studies indicate an elevated risk of non-union.

**Table 1 bioengineering-10-00085-t001:** Studies analyzing the effect of different factors on bone healing.

Factor	Study	Study Design	Bone	Patients (n)	Non-Unions (n)	Follow-Up (yrs)	Result (OR)
Age	[83]	Prospective	Clavicle	245	15	9–10	1.3
	[84]	Prospective	Clavicle	868	53	0.5	-
	[85]	Prospective	Humerus	110	16	1	neg
	[86]	Retrospective	Clavicle	337	19	≥0.5	1.07
	[87]	Retrospective	Humerus	1027	11	1	neg
	[88]	Meta-Analysis	Diaphyseal	38,465	3975		neg
Sex	[88]	Meta-Analysis	Diaphyseal	38,465	3975		neg
Smoking	[89]	Prospective	Tibia	85	9	3	18.46
	[90]	Prospective	Long bone	736	124	1	1.73
	[91]	Prospective	Tibia	647	41	0.5	2.417
	[92]	Retrospective	Clavicle	1196	125	-	3.76
	[93]	Retrospective	Diaphyseal	114	38	≥1	4.14
	[94]	Retrospective	Humerus	2230	231	n.a.	pos
	[95]	Retrospective	Humerus	659	24	0.75	5.34
	[96]	Retrospective (case-control)	Femur	32	32	n.a.	2.29
	[97]	Meta-Analysis	All bones	6356			2.32
	[98]	Meta-Analysis	All bones	39,920			2.5
	[99]	Meta-Analysis	All bones	7516			2.2
	[88]	Meta-Analysis	Diaphyseal	38,465			1.7
Open fracture	[90]	Prospective	Long bone	736	124	1	2.49
	[100]	Retrospective (case-control)	Limb	446	223	0.75	2.71
	[101]	Retrospective	Tibia	486	56	n.a.	4.91
	[97]	Meta-Analysis	All bones	6356			1.95
Displacement	[84]	Prospective	Clavicle	868	53	0.5	neg
	[102]	Prospective	Clavicle	222	15	2	-
	[92]	Prospective	Clavicle	941	125	-	1.17
Diabetes	[88]	Meta-Analysis	Diaphyseal	38,465	3975		1.6
Obesity	[103]	Retrospective (delayed union)	Extremity	147	-	-	3.39
	[104]	Retrospective (osteotomy)	Femur	150	7	1	
	[88]	Meta-Analysis	Diaphyseal	38,465	3975		1.5
Osteoporosis	[105]	Prospective (case-control)	All bones	1498	40	1	neg
	[95]	Retrospective	Humerus	659	24		pos
	[100]	Retrospective (case-control)	Limb	446	223	0.75	3.16
Alcohol	[106]	Prospective	Femur	112	9	4.8 (mean)	neg
	[98]	Meta-Analysis	All bones	39,920			0.97
Drugs							
NSAIDs	[107]	Retrospective (case-control)	All bones	2257	401	1	2.6
	[96]	Retrospective (case-control)	Femur	32	32	n.a.	10.74
	[100]	Retrospective (case-control)	Limb	446	223	0.75	2.04
	[95]	Retrospective	Humerus	659	24	0.75	2.51
Steroids	[91]	Prospective	Tibia	647	41	2	neg
	[108]	Retrospective (pediatric)	All bones	237,033	2003	1	pos
PPIs	[109]	Retrospective (case-control)	Femur and tibia	254	12	n.a.	4.5

## Data Availability

Original data are available on personal request from the corresponding author.

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
