# Peer review of "Bone Healing Gone Wrong: Pathological Fracture Healing and Non-Unions—Overview of Basic and Clinical Aspects and Systematic Review of Risk Factors"

_bioengineering, 2023, doi:10.3390/bioengineering10010085_

Round 1

Reviewer 1 Report

-

Author Response

According to the comment of the reviewer, the entire manuscript has been carefully checked by a native speaker for overall language errors, typos and grammar errors.

Reviewer 2 Report

Bone healing after fracture happens to fail in 2-10% of all fractures with devastating effects for both the patient and health care system. Underlying reasons for this failure are manifold and not always known. This review aims to describe the epidemiology, the cellular components involved in impaired bone healing, the risk factors for the development of a non-union, therapeutical prospects and the applicability of biomarkers for impaired bone healing and development of a non-union.

The review is timely, however not overly original as there are several good and extended systematic reviews published discussing the topic of failed bone healing.

General points

-          Chapter 5: Risk factors

Here, the immune system might be included as immune dysregulation is a risk factor for impaired bone healing.

-          The chapter “therapeutic options” ranges too short.

If discussing non-union therapies, biomaterial/scaffold (either cell-free or cell-combined ) supported treatments need to be included.

In addition, adipose tissue as a source of MPCs should not be ignored as it is even more feasible and easy to use for extraction of MPCs as the bone marrow source.

What about the usage of genetically modified MSCs as a novel therapeutic option (see review by Freitas et al. IJMS 2019; https://doi.org/10.3390/ijms20143430) ?

-          The same goes for chapter 7: “Biomarkers”.

This reviewer misses the description of various blood cell types, which can serve as biomarker candidates for prediction of bone regenerative capacities. These include Myeloid-derived suppressor cells and B cells.

Author Response

Here, the immune system might be included as immune dysregulation is a risk factor for impaired bone healing.

We thank the reviewer for this helpful comment. Accordingly, we have added a novel paragraph in chapter 5, which describes immune dysregulation as a potential risk factor for impaired bone healing. This paragraph reads as follows:

“Immune dysregulation has been hypothesized to impair fracture healing. While the initial inflammatory phase and local immunosuppression are timely resolved, later stages require immune cells to guide MSCs differentiation and activity [112]. In all healing stages, immune homeostasis is controlled by CD4+, CD8+ and regulatory T cells [113]. Accordingly, a T cell defect or compromised immune system as seen in HIV patients with deranged levels of TNF-α, is associated with impaired fracture healing [114]. While the complex interplay of inflammation and the immune system is further affected by diseases, such as diabetes or habits like smoking or even aging itself, larger prospective trials on the effects of immune dysregulation on fracture healing are missing [115].”

(See page 10, lines 350-358)

-          The chapter “therapeutic options” ranges too short.

If discussing non-union therapies, biomaterial/scaffold (either cell-free or cell-combined ) supported treatments need to be included.

We thank the reviewer for this comment. Accordingly, we have added a novel paragraph in chapter 5, which discusses biomaterial/scaffold-supported treatments. This paragraph reads as follows:

Biomaterial-supported therapies for bone regeneration include a magnitude of different approaches. Among them, bone graft substitutes, a combination with bio-active molecules and cell-based supplements with osteoprogenitors have been applied in vivo [42]. Bone graft substitutes include autologous and allogenous bone substitutes. While the former are limited in their availability from the iliac crest, the latter are usually preserved in a freezing medium and, in consequence, lack some of the osteogenic properties and the mechanical stability of autografts. The cryopreservation process, however, avoids a donor-site immune response [137]. Despite these limitations, the implantation of autograft and allograft substitutes remains the current gold standard for the treatment of larger bone defects [42]. The addition of bioactive molecules, such as growth factors like BMP-2 or -7 and PDGF or the peptide P-15 has emerged in the last decades. This is demonstrated by an FDA-approval for all of them [138]. After dose-dependent inflammatory side effects limited the use of BMP-2 therapy [139], results from new clinical trials on BMP-2 are expected soon (NCT02924571, NCT05065684, NCT05238740), while clinical studies on P-15 in spinal deformity bone grafting (NCT05038527, recruiting in 02/2022) and lumbar fusion (NCT03438747, active, not recruiting in 02/2022) are ongoing. Cell-based supplements usually contain bone marrow stromal cells (BMSCs), adipose-derived mesenchymal stem cells (AMSCs), or periosteum-derived stem cells [42]. Particularly for larger defects, an additional and easily applicable source for MSCs is the adipose tissue, that can be efficiently harvested for the isolation of AMSCs [140]. BMSCs are widely used in clinical trials, either with or without the addition of BMP-2, and injected into a medium or seeded into a matrix before applied in a bone defect. A clinical trial on intervertebral disc defect repair with BMSCs in a gelatin sponge is currently recruiting (NCT03002207, 10/2021). A recent meta-analysis concluded that a scaffold positively impacts the healing rate of cell-based scaffolds [141]. However, prospective randomized controlled trial long-term follow-up on scaffold-based BMSC treatment of non-unions is still missing.

(See page 11, lines 388-413)

In addition, adipose tissue as a source of MPCs should not be ignored as it is even more feasible and easy to use for extraction of MPCs as the bone marrow source.

According to the comment of the reviewer, we have added this aspect in the new chapter on biomaterials, which reads as follows: “Particularly for larger defects, an additional and easily applicable source for MSCs is the adipose tissue, that can be efficiently harvested for the isolation of AMSCs [140].”.

(See page 11, lines 405-407)

What about the usage of genetically modified MSCs as a novel therapeutic option (see review by Freitas et al. IJMS 2019; https://doi.org/10.3390/ijms20143430) ?

According to the comment of the reviewer, we have included a novel paragraph into our revised manuscript, which reads as follows:

“Apart from embedding freshly isolated MSCs into a scaffold, they can be further genetically modified in order to improve their osteogenic properties or proliferative potential. This can be achieved by viral-based gene-delivery systems using a gene-activated matrix, genome editing via the CRISPR/Cas9 technique or engineered extracellular vesicles (EVs) [142].The latter ones are also a cell-free alternative. In fact, MSC-derived EVs alone have been shown to improve fracture healing in rodent models [44,46].

(See page 11, lines 413-419)

-          The same goes for chapter 7: “Biomarkers”.

This reviewer misses the description of various blood cell types, which can serve as biomarker candidates for prediction of bone regenerative capacities. These include Myeloid-derived suppressor cells and B cells.

We agree with the reviewer that a description of cellular components which might potentially serve as biomarkers for bone regeneration is missing. Accordingly, we have added a novel paragraph in the revised manuscript version, which reads as follows:

“However, pre-clinical studies are promising. Both cytokines like IL-10 and myeloid-derived suppressor cells (MDSCs) were significantly elevated in blood one week after treatment, prior to poor bone healing outcome in a rat critical size defect model. Interestingly, blood B cell levels were positively correlated to bone volume as early as one week after treatment, indicating an early predictor for successful healing. Given the importance of B cell maturation for bone healing, these cells might be promising markers for bone repair, although clinical trials are missing yet [151]. Moreover, the local bone marrow showed consistent MDSC levels with the blood immune response profile 12 weeks after the treatment. This promising cell population secretes various anti-inflammatory factors like IL-10 and TGF-β, inhibits the immune response and is expanded in aging [152,153].”

(See page 12, lines 441-450)

Reviewer 3 Report

The authors present here a very interesting systematic review dealing with repair defects and fracture healing. The study focuses on the risk factors associated with bone non-union formation.

The manuscript is particularly well written, the study is judiciously conducted and the results are correctly interpreted.

However, we can only regret not taking into account the osteocyte as a mechanosensitive cell orchestrating and directing the targeted bone remdodeling.

Author Response

We thank the reviewer for these encouraging words and positive assessment of our work.

However, we can only regret not taking into account the osteocyte as a mechanosensitive cell orchestrating and directing the targeted bone remdodeling.

We thank the reviewer for this suggestion and agree that the osteocyte as orchestrator of new bone formation needs to be added to this review. Accordingly, we have added the following paragraph to the revised version of our manuscript:

“4.5. Osteocytes

Osteocytes are mechanosensitive cells sensing the deformation of surrounding tissue via flow-induced shear stress on their surface. Their exact function and the role of the lacuno-canalicular network has been outlined in detail elsewhere [64]. The mesenchymal-derived cell that is most abundant in the adult bone coordinates osteoblast and osteoclast activity and arranges the adaption to environmental changes via release of various molecules [65,66]. While their exact role in fracture healing is largely unknown, their involvement in aging, and senescent phenotype has been linked to an age-dependent skeletal decline [67,68].

(See page 6, lines 219-227)

Reviewer 4 Report

This review summarizes the epidemiological aspects of broken bones and the socio-conomic consequences of occurring non-union. The cellular key players in the (non-) physiological healing process are unrolled and their disturbance briefly discussed. These cellular insights are followed by a systematic review of the risk factors for non-union. Finally, therapeutic options are outlined and the application of potential biomarkers are discussed.

Specific comments:

Line 64 ff. Figure 1. PRISMA flowchart of data extraction, in the screening process, case reports and animal studies were removed, why did you list their exclusion again in the later step of eligibility testing?

Line 99 ff. Epidemiology – what about other countries? Only data for the US are given. How about other Is there a difference in the continents? Since the scope of the article is summarizing epidemiological aspects, more data would be desirable. 

Line 131. Figure 2. Cellular components of contributing to (impaired) bone healing. Is this the summary of the review? Please mark more clearly, or provide literature sources within the legend. 

Line 157 ff. Clinical trials with MSC transplantation – please give more information about the use of autologous MSC and the date of the study start, so that the reader is informed whether this is an actual or an ancient approach.

Line 185 Osteoprogenitors and MSC:  It would be interesting to know if there is further research on the effects of MSC-derived Evs on bone healing.

Figure 3. Risk factors for bone non-union formation. The symbols in particular for retrospective are not clearly designated to: obesity/osteoporosis/alcohol? Please explain all abbreviations used in the legend: PPIs? Proton pump inhibitors.

Line 338 ff. initial positive studies were published in 1994 and 1997 – is there anything more recently published? Is this still a method which is used?

Line 347 – [115] published in 2013 – anything new since then?

Chapter 6: Therapeutic options:  In general, it would be desirable to have more information on the timeliness of the method and whether studies /RCTs are currently underway on it.

The same is true for the biomarker in chapter 7 - the reader does not know if TGF-b1 is still a candidate biomarker, like IL-10 or annexin A 3 described in the recent 2021 and 2022 studies.

Author Response

Line 64 ff. Figure 1. PRISMA flowchart of data extraction, in the screening process, case reports and animal studies were removed, why did you list their exclusion again in the later step of eligibility testing?

We apologize for the mistake and corrected accordingly that animal studies were just removed in the first screening step. Figure 1 has been adjusted in the revised version of our manuscript.

Line 99 ff. Epidemiology – what about other countries? Only data for the US are given. How about other Is there a difference in the continents? Since the scope of the article is summarizing epidemiological aspects, more data would be desirable.

We corrected the lack of data from other countries and added: “While worldwide data is limited, the global incidence of non-union is estimated around 5-10%, depending on the fracture location and country with China reporting a non-union rate of 4.7% in tibial fractures, and Singapore reporting a 42.7% non-union rate (just open Gustilo-Anderson IIIB tibial fractures were included). Smaller and unrepresentative studies in Turkey and Egypt reported lower rates in tibial non-unions of 1.4 and 3.3%, respectively[19].”.

(See page 3, lines 99-104)

Line 131. Figure 2. Cellular components of contributing to (impaired) bone healing. Is this the summary of the review? Please mark more clearly, or provide literature sources within the legend.

We thank the reviewer for pointing out the missing clarity and added the literature supporting each statement in the legend of Fig. 2.

Line 157 ff. Clinical trials with MSC transplantation – please give more information about the use of autologous MSC and the date of the study start, so that the reader is informed whether this is an actual or an ancient approach.

We thank the reviewer for pointing out the missing clarity. We added the information about all clinical trials, and likewise added “Nonetheless, bone grafting with either autograft or allograft containing (undefined proportions and quality of) MSCs remains the current gold standard in patients with incomplete bone healing [42].” to inform the reader of the actual approach for incomplete fracture healing.

(See page 5, lines 164-166)

Line 185 Osteoprogenitors and MSC:  It would be interesting to know if there is further research on the effects of MSC-derived Evs on bone healing.

We thank the reviewer for this idea and added “A cell-free approach in treating bone defects are MSC-derived extracellular vesicles (MSC-EVs) [43]. These may contain monocyte chemotactic protein 1 (MCP-1) and several angiogenic factors that accelerate fracture healing in rodent models [44–46].” to chapter 4.1 and to chapter 6: “The latter ones are also a cell-free alternative. In fact, MSC-derived EVs alone have been shown to improve fracture healing in rodent models [44,46].”

(See page 5, lines 167-169)

Figure 3. Risk factors for bone non-union formation. The symbols in particular for retrospective are not clearly designated to: obesity/osteoporosis/alcohol? Please explain all abbreviations used in the legend: PPIs? Proton pump inhibitors.

We thank the reviewer for pointing this out. Accordingly, bars have been added to Fig. 3 to clearly indicate which symbol belongs to which column. Likewise, PPI is explained in the legend of Fig. 3 now.

Line 338 ff. initial positive studies were published in 1994 and 1997 – is there anything more recently published? Is this still a method which is used?

We thank the reviewer for pointing out the missing clarity and added “Today, although there are several animal studies with favorable data for specific indications like bone-tendon healing [128], LIPUS is generally not used in the clinic except for rare and individual treatment if surgery is not feasible [129].”.

(See page 10, lines 379-382)

Line 347 – [115] published in 2013 – anything new since then?

Yes. We added “Preliminary studies are promising [132] and both in vitro and in vivo trials encourage the use of PEMF in osteoporosis and delayed union [133–136]. A clinical trial in distal radius fractures is currently ongoing (“not yet recruiting” in 02/2020).”.

(See page 11, lines 384-387)

Chapter 6: Therapeutic options:  In general, it would be desirable to have more information on the timeliness of the method and whether studies /RCTs are currently underway on it.

For LIPUS, a clarifying sentence has been added. For PEMF, “A clinical trial in distal radius fractures is currently ongoing (“not yet recruiting” in 02/2020).” has been added.

(See page 11, lines 386-387)

Additionally, for all clinical trials in chapter 6, the current status has been added to inform the reader on the timeliness on each method.

The same is true for the biomarker in chapter 7 - the reader does not know if TGF-b1 is still a candidate biomarker, like IL-10 or annexin A 3 described in the recent 2021 and 2022 studies.

We apologize for the missing clarity and added “Newer data suggests that TGF-β1 impairs MSC-mediated bone regeneration, mechanistically through BMP2 inhibition, but its role as biomarker has not been explored lately with no clinical trials ongoing [150].” to ensure the reader knows the current status of TGF-b1 as a potential biomarker is on hold. To further inform the reader on the status on each potential biomarker, we have added the publication date on each of the findings. However, no clinical trials for any of these markers are ongoing.

(See page 11-12, lines 435-437)

Round 2

Reviewer 2 Report

The authors have addressed my concerns very well and the manuscript has improved significantly.